# Effect of Oxidation and Crystallization on Pitting Initiation Behavior of Fe-based Amorphous Coatings

**Haoran Zhang** [1], **Shanlin Wang** [1,*], **Hongxiang Li** [2], **Shuaixing Wang** [3] and **Yuhua Chen** [1,*]

1 Jiangxi Key Laboratory of Forming and Joining Technology for Aviation Components, Nanchang Hangkong University, Nanchang 330063, China; 1903085204125@stu.nchu.edu.cn
2 State Key Laboratory for Advanced Metals and Materials, University of Science and Technology Beijing, Beijing 100083, China; hxli@skl.ustb.edu.cn
3 School of Materials Science and Engineering, Nanchang Hangkong University, Nanchang 330063, China; wsxxpq@126.com
* Correspondence: slwang70518@nchu.edu.cn (S.W.); yuhuachen@nchu.edu.cn (Y.C.); Tel.: +86-13755603426 (S.W.)

**Abstract:** Fe-based amorphous coatings are typically fabricated by high-velocity oxygen-fuel spraying using industrial raw materials. The bonding mode between the coating particles and the corrosion mechanism of the coating in the chloride-rich environment were studied. The results indicate that some fine crystallites such as $\alpha$-Fe and $Fe_3C$ tend to precipitate from the amorphous matrix as the kerosene flow rate increases or the travel speed of spraying gun decreases. Moreover, some precipitates of the $(Cr, Fe)_2O_3$ nanocrystal were detected in the metallurgical interfaces of the amorphous coating. The relationship among the amorphous volume fraction, porosity, and spraying parameters, such as the kerosene flow rate and the travel speed of the spray gun, were established. Due to an oxidation effect during spraying process, atomic diffusion, crystallite precipitation and regional depletion of Cr occur in the area along the pre-deposited side near the metallurgical bonding interface, leading to the initiation of pitting. A model of pitting initiation and expansion of Fe-based amorphous coatings is proposed in this paper.

**Keywords:** Fe-based amorphous coating; forming; pitting corrosion; high-velocity oxygen-fuel spraying

## 1. Introduction

Owing to their excellent corrosion resistance and low material cost, Fe-based amorphous coatings offer considerable potential for industrial application in fields such as the military, nuclear power, oil and gas, and manufacturing. Farmer [1,2] attempted to apply Fe-based amorphous coatings on containers used for the transportation, aging, and disposal of spent nuclear fuel and high-level radioactive wastes. The in-transmission neutron absorption cross section for thermal neutrons of $Fe_{49.7}Cr_{17.7}Mn_{1.9}Mo_{7.4}W_{1.6}B_{15.2}C_{3.8}Si_{2.4}$ (SAM2X5) with a high boron content is four times greater than that of borated stainless steel, and twice as that of nickel-based alloy (C–4) with added Gd. In addition, SAM2X5 amorphous coatings are being used as corrosion-resistant anti-skid decking for ships. No significant irradiation damage phenomena appear in the $Fe_{80}Si_{7.43}B_{12.57}$ metallic glass, while blistering, flaking, and other damage occurs on the surface of the metallic W in an environment subjected to $H^+$ irradiation [3]. The Fe-based amorphous coating with the composition of $Fe_{54.2}Cr_{18.3}Mo_{13.7}Mn_2W_6B_{3.3}C_{1.1}Si_{1.4}$ (wt.%) used on marine pump impellers has an erosion rate that is two to three times that of SUS304, and it is believed to enhance the lifetime of pump impellers that need to work in sand-containing seawater [4]. The corrosion rate of the Fe-based amorphous coating is 1/1000 that of 20G steel in the artificial simulation environment of a power-plant boiler [5], which demonstrates that the

amorphous coating has an excellent anti-corrosion performance with controllable moderate investment cost that can promote the engineering application of waste heat utilization in power plants.

The corrosion resistance of Fe-based amorphous coatings has improved considerably in recent decades. In 1996, Otsubo [6] prepared FeCr-based amorphous coatings that exhibit corrosion resistance superior to that of SUS316 when being exposed to environments rich in sulfuric acid and hydrochloric acid. Furthermore, compared to the aggressive attack on SUS316, the amorphous coatings undergo neither general nor pitting corrosion when immersed in 1 mol/L HCl solution or 6% $FeCl_3$ solution containing 0.05 mol/L HCl. Recently, most studies have focused on elements related to corrosion resistance, such as spraying technologies, parameters, elements, porosity, and microstructure, etc. [7]. The passive current densities of the Fe–16%Cr–30%Mo–(C, B, P) amorphous coatings sprayed by the high-velocity oxygen-fuel (HVOF) spraying and atmospheric plasma spraying (APS) processes are close each other and significantly low compared to those of SUS316L coating in a 1 mol/L HCl solution [8]. The increase of kerosene and oxygen flow leads to the decrease in porosity and amorphous fraction of the coatings, and the optimal spraying parameters improve the corrosion resistance of the amorphous coatings due to the proper ratio of porosity and amorphous fraction in acidic solutions [9]. The corrosion resistance deteriorates with the increasing precipitation of the crystalline phases, and the pits are always initiated at the boundaries around the carbides where a Cr- and Mo-depleted zone exists [10]. While the $Fe_3Si$ nanocrystallites and $FeCu_4$ phases decrease the corrosion rate in a 0.5 mol/L HCl solution, the formation of a composite layer consisting of $Fe_3O_4$ and $SiO_2$ results in some protective properties [11]. Due to the precipitation of the $\alpha$-Fe phase, the corrosion rate is accelerated evidently for the galvanic effect [12].

As known, the homogeneity of the microstructure and chemical composition results in an excellent corrosion resistance [13,14]. However, the destruction of the homogeneity is inevitable to the formation of cavities, interfaces between lamellar structures, oxides, even crystallization in Fe-based amorphous coating, which results in the initiation of pitting corrosion in the Fe-based amorphous coating in a corrosive environment [15–17]. Gostin [18] found that a pitting-like process occurred in glassy alloys with a high concentration of C as the initial breakdown of the passive film caused sudden direct exposure of the alloy surface to the electrolyte after local rupturing of the C-rich layer by growing $CO_2$ bubbles. Zhang [19] suggested that the formation of $Fe_3O_4$ nanocrystals along the interface of SUS316 and the amorphous matrix leads to a galvanic effect which triggers the pitting corrosion. Wang [15] also reported that the pitting corrosion of amorphous metallic coatings is related to oxides of Fe, Cr, and Mo. $Cr_2O_3$ inclusions induce pitting initiation for amorphous metallic coatings in a chloride solution. Liu et al. [16,20] proposed that the pitting corrosion of Fe-based amorphous metallic coatings originates from the amorphous matrix areas, which are narrow regions deficient in Cr, with a width of approximately 100 nm near the interface owing to the action of the oxides. However, the initiation and progress of pitting corrosion in Fe-based amorphous coatings have not been studied extensively. Therefore, in this study, the formation, microstructure, and pitting initiation are investigated for Fe-based amorphous coatings with industrial raw materials.

## 2. Methods

Fe-based amorphous powders with the mass composition of $Fe_{44.72}Co_{8.57}Cr_{14.95}Mo_{26.9}C_{3.2}B_{1.28}Y_{3.01}$ were prepared via a commercial gas atomization technique using industrial raw materials. Particles with a diameter of 50–70 μm were screened by a sieve to ensure the fluidity of the powders during the spraying process. A low-alloy carbon steel plate with a dimension of 100 mm × 100 mm × 5 mm was degreased in acetone and sandblasted to secure good bonding between the coating layer and the substrate. The fuel used (Lijia, Zhengzhou, China) was aviation kerosene with a molecular weight of 154, relative density (A ratio of density to water) of 0.825, and combustion heat of 45.1 J/g. A spray gun (Lijia, Zhengzhou, China) with a length of 4 inches (10.16 cm) ensures full

combustion with pure oxygen as a catalyst. The coatings were produced by high-velocity oxy-fuel spraying (HVOF, HV-80-JP, Lijia, Zhengzhou, China) with the spraying experimental condition given in Table 1.

**Table 1.** Spraying parameters of HVOF process.

| No. | Oxygen Flow Rate (m³/h) | Kerosene Flow Rate (L/h) | Feed Rate (g/min) | Spraying Distance (mm) | Speed of Gun (m/min) |
|-----|-----|-----|-----|-----|-----|
| 1 | 50 | 32 | 72 | 380 | 10 |
| 2 | 50 | 29 | 72 | 380 | 10 |
| 3 | 50 | 26 | 72 | 380 | 10 |
| 4 | 50 | 23 | 72 | 380 | 10 |
| 5 | 42 | 15 | 50 | 300 | 10 |
| 6 | 42 | 15 | 50 | 300 | 8 |
| 7 | 42 | 15 | 50 | 300 | 6 |
| 8 | 42 | 15 | 50 | 300 | 4 |

The crystalline phases in the coatings were identified by x-ray diffraction (XRD, Philip X'Pert diffractometer, PANalytical B.V., Netherlands) with Cu radiation (Cu-K$\alpha$, $\lambda$ = 0.1541 nm). The thermal stability was investigated by a differential scanning calorimeter (DSC, Pyris Diamond DSC, Perkin Elmer, Akron, OH, USA) with a heating rate of 20 °C/min. Electrochemical measurements were conducted using a CHI650 potentiostat (CH Instruments Ins., Austin, TX, USA). The electrochemical cell contained a central inlet for the working electrode, Luggin probe connected with a saturated calomel reference electrode (SCE, KCl, Beijing Instrument Electric Technology, Beijing, China), and graphite counter electrode (Beijing Instrument Electric Technology, Beijing, China). The potentiodynamic polarization curves were measured with a potential sweep rate of 1 mV/s in different concentrations of HCl and NaCl solutions, respectively, open to air at 293K after immersion for 30 min. An immersion test was performed in 6 mol/L NaCl solution to assess the pitting corrosion. The surface morphologies were observed after spraying and etching with a scanning electron microscope (SEM, SV3400, SMC, Tokyo, Japan), and the interface microstructures was detected by transmission electron microscopy (TEM, TecnaiG2 F20 S-TWIN TMP, FEI, Hillsboro, OR, USA).

## 3. Results

### 3.1. Formation of Amorphous Coatings

The morphology of the spraying powder is shown in Figure 1a, which indicates that the powders were mostly spherical with smooth surfaces that imply favorable fluidity. Figure 1b shows the bright field image (BFI) and selected area electron diffraction pattern (SAED) of the spray powders detected by TEM. The diffraction halo ring from the SAED pattern suggests that the powders had a completely amorphous structure.

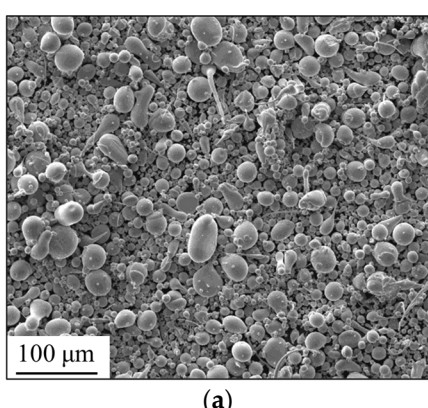

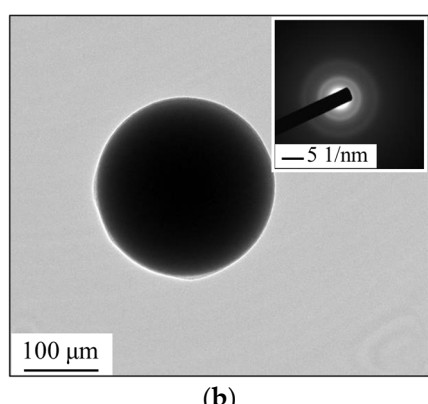

(**a**)  (**b**)

**Figure 1.** Morphologies of Fe-based amorphous powders: (**a**) SEM micro morphology, (**b**) SAED pattern of powders.

The morphologies of the as-sprayed surface and cross section of the amorphous coatings are shown in Figures 2 and 3. It can be observed that the cross section of the coatings exhibited a pancake-like morphology, implying that most of the spray particles underwent considerable plastic deformation when they impacted the substrate surface. The lamellar coatings with uniform thickness were well bonded to the substrate, as shown in Figure 2b–h and Figure 3b–h. While some pores could be observed around the semi-fused particles, which themselves remained spherical due to insufficient melting. With the increase in the kerosene flow rate, the spray particles tended to spread out into a more evident lamellar shape, as shown in Figure 2. The number and size of pores decreased with an increase in the degree of extension of the particles, and the coating tended to be dense. However, the change in the lamellar structure was not evident with the variation in the travel speed of the spray gun, as shown in Figure 3.

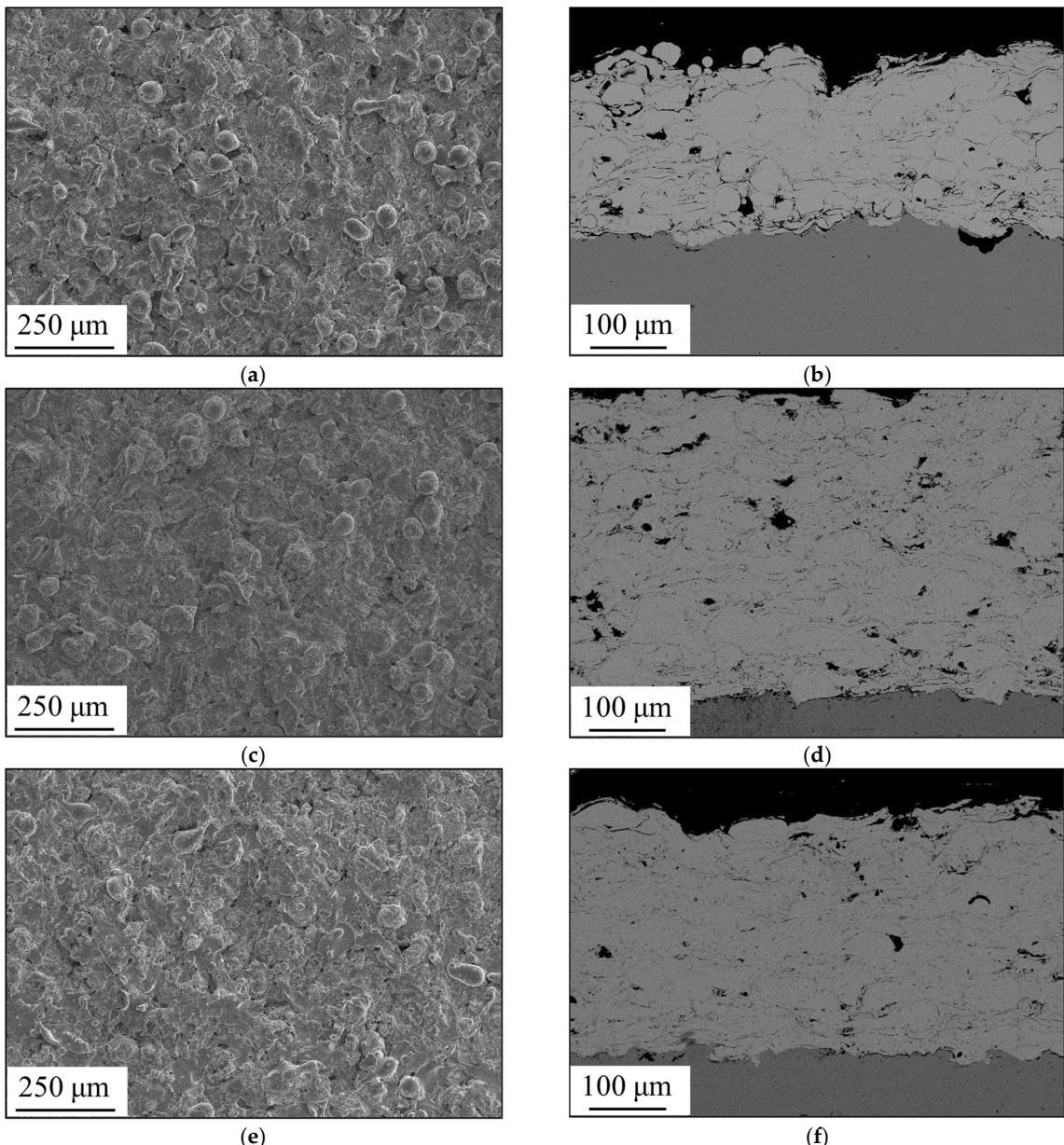

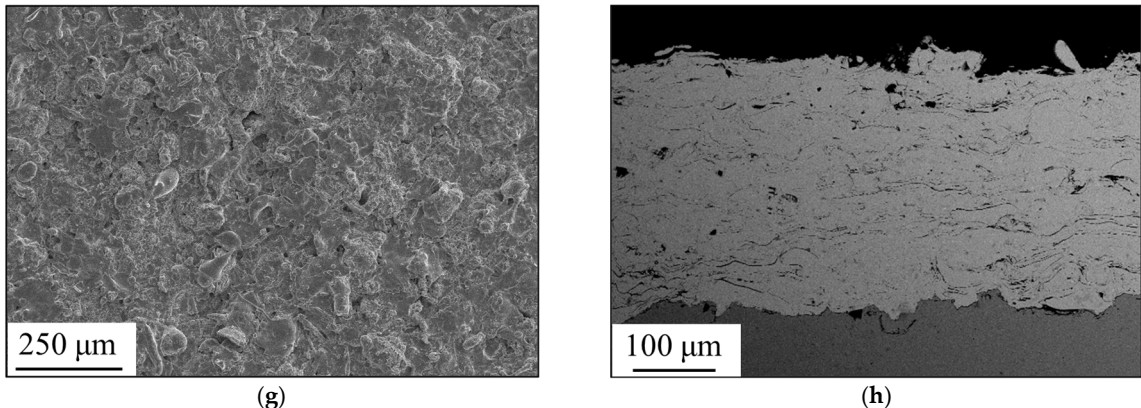

(**g**)  (**h**)

**Figure 2.** Morphologies of Fe-based amorphous coatings fabricated with different kerosene flow rate, (**a**) and (**b**) 23 L/h, (**c**) and (**d**) 26 L/h, (**e**) and (**f**) 29 L/h, (**g**) and (**h**) 32 L/h.

(**a**)  (**b**)

(**c**)  (**d**)

(**e**)  (**f**)

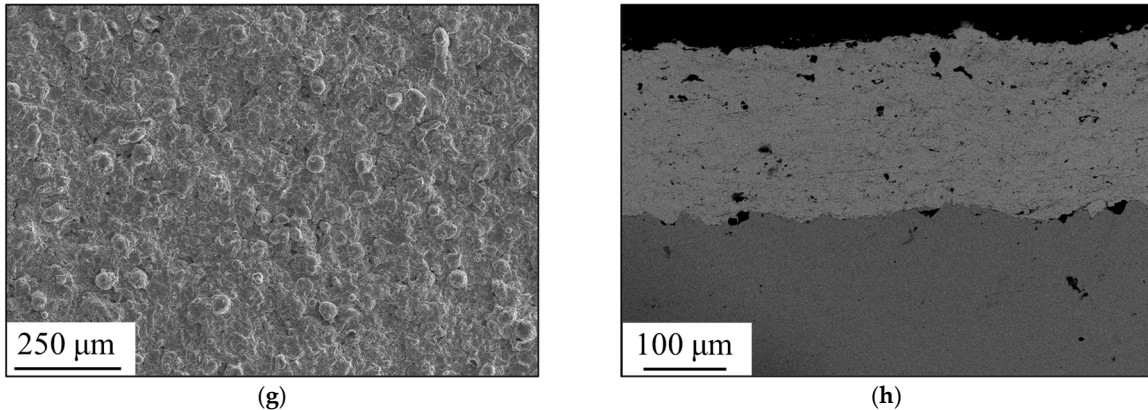

**Figure 3.** Morphologies of Fe-based amorphous coatings fabricated with different travel speed of spraying gun, (**a**) and (**b**) 4 m/min, (**c**) and (**d**) 6 m/min, (**e**) and (**f**) 8 m/min, (**g**) and (**h**) 10 m/min.

The XRD pattern of the powder and the as-sprayed coatings are displayed in Figure 4, which shows the similarity in XRD patterns between the amorphous powder and the coatings. Some crystalline diffraction peaks overlapped a broad diffraction halo at the angle of $2\theta = 40°\sim50°$, suggesting that the coatings were partially nanocrystallized or crystallized. When the kerosene flow rate was 23 L/h, some FeO peaks could be detected from the amorphous coating. When the kerosene flow rate increased to 26 L/h, no crystalline diffraction peak could be observed except the diffraction halo peak. With the continuous increase in the kerosene flow rate, an $\alpha$-Fe peak occurred in these coatings, as shown in Figure 4. With the decrease in the travel speed of gun, the amorphous coating was gradually crystallized, and some crystalline phases of $\alpha$-Fe, FeO, and $Fe_3C$ are precipitated successively.

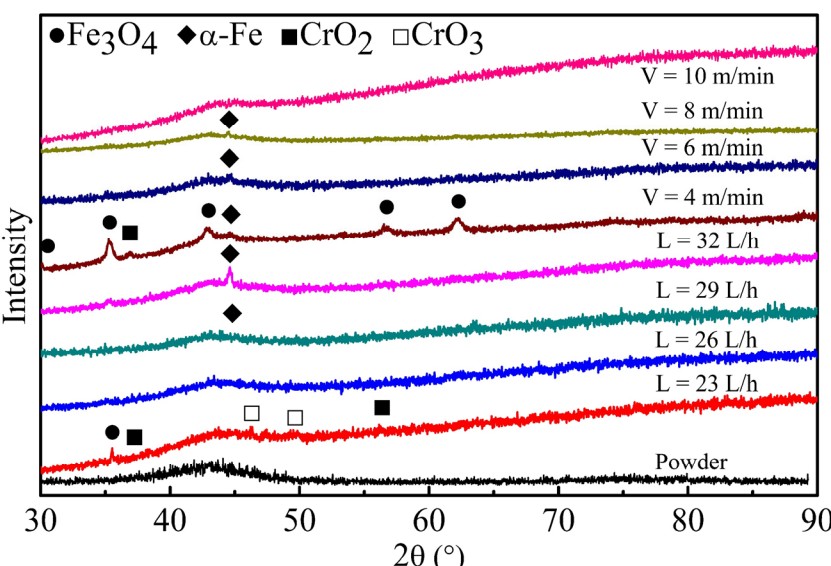

**Figure 4.** XRD patterns of Fe-based amorphous coatings with different spraying parameters.

Figure 5 shows the DSC curves of the amorphous coatings sprayed with different kerosene flow rates and travel speed of gun. The amorphous proportion (a) of the coatings can be calculated based on the crystallizing enthalpies of the coatings ($\Delta H_c$) and amorphous powders ($\Delta H_0$) according to the function a = $\Delta H_c/\Delta H_0$ [21]. The coatings fabricated at a kerosene flow rate of 26 L/h had exothermic enthalpies ($\Delta H$) like that of amorphous powders, and an ultra-high amorphous proportion (99.41%), indicating that no serious crystallization occurred. A decrease in exothermic enthalpy implied an increase in the degree of crystallization during spray-coating. However, with the decrease in the travel

speed of the gun, the amorphous proportion of the fabricated coatings decreased from 99.60% at 10 m/min to 32.21% at 4 m/min due to the partial crystallization, as shown in Figure 5. In evidence, the quantified exothermic enthalpy for all the amorphous coatings matched the crystallization peaks in the XRD results. The variation of the amorphous fraction and porosity with respect to the kerosene flow rate and travel speed is plotted in Figure 6. With the increase in travel speed, the amorphous volume fraction increased continuously up to 99%. Similarly, the porosity decreased continuously with the increase in kerosene flow rate and the decrease in travel speed. The lowest porosity was lower than 1.0%, while the amorphous volume fraction was up to 80 at.%.

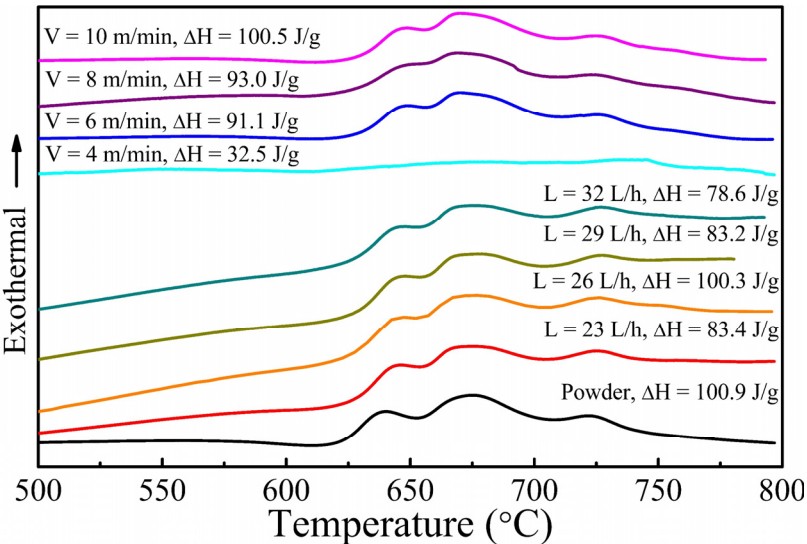

**Figure 5.** DSC curves of Fe-based amorphous coatings with different spraying parameters.

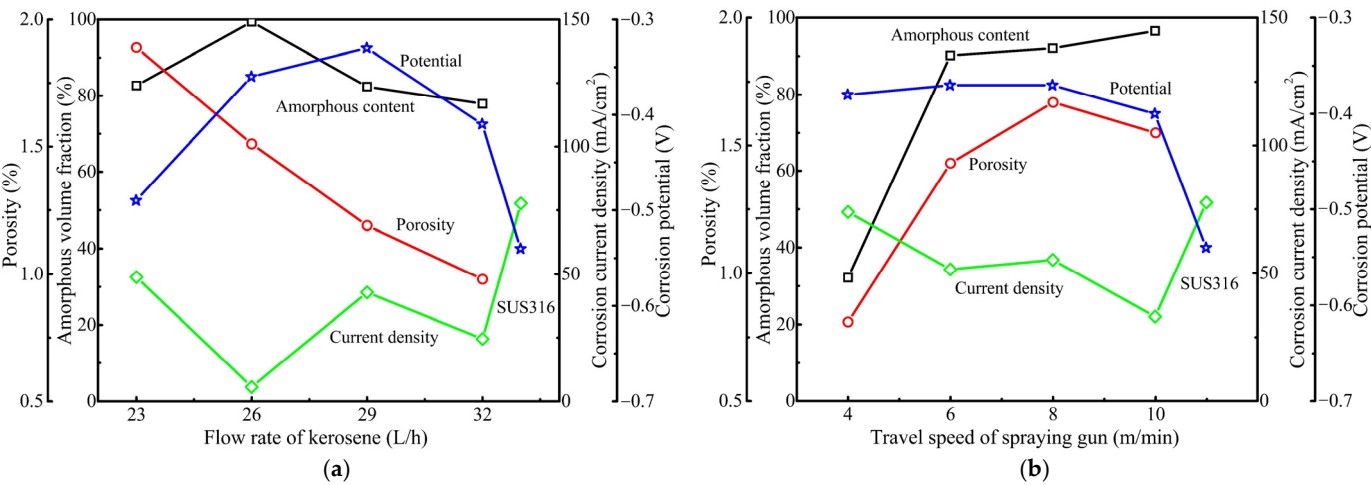

**Figure 6.** Relationship of amorphous volume fraction, porosity, corrosion potential, corrosion current density and spraying parameters of amorphous coatings, (**a**) kerosene flow rate, (**b**) travel speed.

### 3.2. Corrosion Resistance of Amorphous Coatings

The potentiodynamic polarization characteristics of amorphous coatings were investigated in 1 mol/L HCl as shown in Figure 7. Obviously, all amorphous coatings exhibited more positive corrosion potential, less corrosion current, and wider passivation region than SUS316 stainless steel, which indicated that all amorphous coatings present superior corrosion resistance in the HCl solution. With the increase in the kerosene flow rate, the corrosion current density of the fabricated coatings decreased to a minimum (5.62 mA/cm$^2$) at 26 L/h, and then increased gradually. Initially, the corrosion potential

increased to the maximum (−0.33 V) at 29 L/h, indicating that the corrosion tendency decreased to the lowest, and then decreased gradually. In terms of both corrosion current and corrosion potential, the corrosion resistance of the coating reaches its maximum with the increase of heat input and then weakens. As the spray gun travel speed increases, the corrosion current density decreases generally, but the corrosion potential remains nearly constant. The tendency of corrosion current density and corrosion potential for all amorphous coatings are plotted in Figure 6. All the amorphous coatings presented lower corrosion current density and more positive corrosion potential than SUS316.

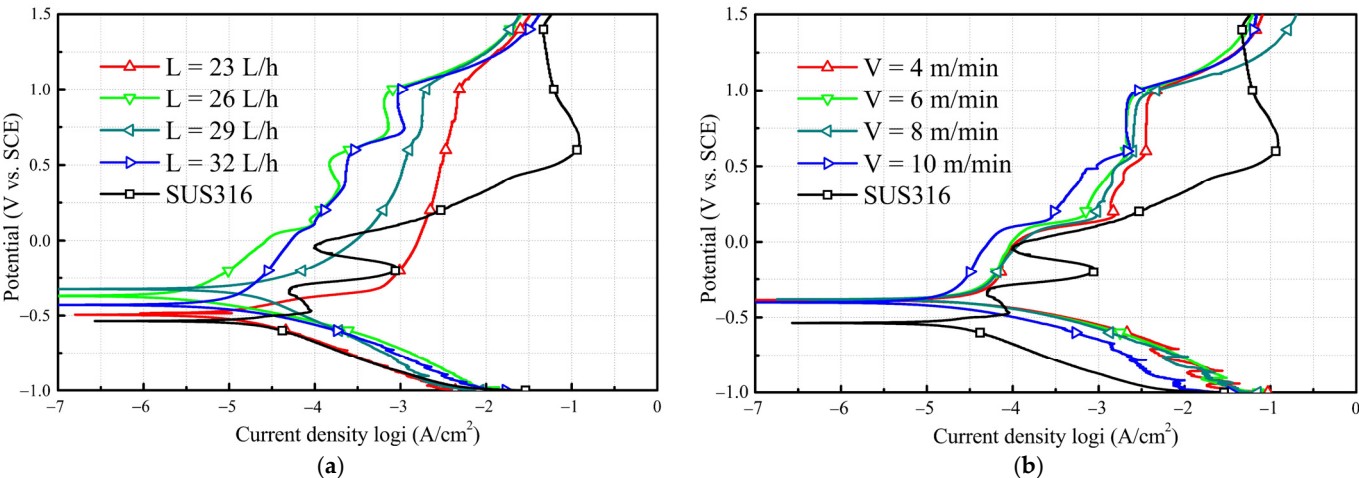

**Figure 7.** Potentiodynamic polarization of Fe-based amorphous coatings in 1 mol/L HCl, (**a**) kerosene flow rate, (**b**) travel speed.

Figure 8 shows the potentiodynamic polarization curves of amorphous coating fabricated with 26 L/h kerosene in HCl solutions with different concentrations. It could be observed that the amorphous coating exhibited a spontaneous passivation with low passive current density and wide passivation region without the peaks caused by the sudden increase in corrosion current in the $Cl^-$ solution, which suggests the amorphous coating possessed a prominent ability to resist localized corrosion [22]. As the $Cl^-$ content increased, the amorphous coating showed a similar polarization characteristic except for an increase in the corrosion current density. The corrosion current density increased more than 20 times when the HCl concentration increased from 0.5 to 4 mol/L; that is, the corrosion current density was from 0.8 mA/cm$^2$ in 0.5 mol/L HCl solution to 18.6 mA/cm$^2$ in 4 mol/L HCl solution. When the HCl concentration was less than 1 mol/L, the amorphous coating presented a smooth surface without obvious corrosive products and pits after the polarization test (Figure 9). When the HCl concentration exceeded 2 mol/L, some pits could be observed on the corrosive surface. Moreover, when the HCl concentration exceeded to 4 mol/L, many pits were formed on the surface, and many 'dried riverbed' cracks occurred on the surface, as shown in Figure 9d. Similar cracks were reported in the literatures [23,24], and this implies that thick corrosive products were precipitated on the specimen surface. In the initial stage of the immersion test, obvious cracks appeared on the outer surface of SUS316 in the chloride solution. Subsequently, the number, width, and depth of the cracks increased rapidly, and the corrosion product layer took on a loose, porous honeycomb structure with low adhesion, making it easy to fall off [25,26].

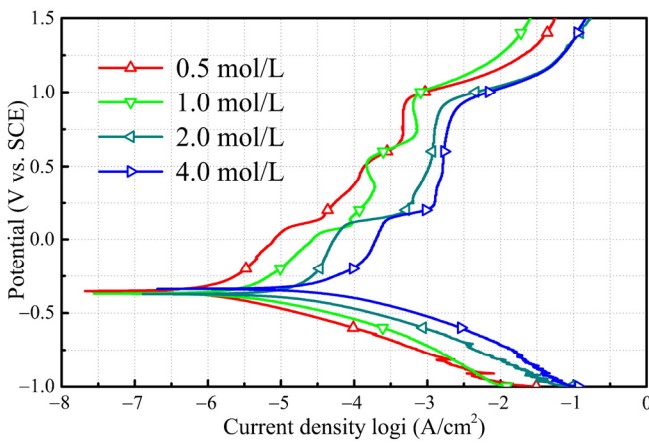

**Figure 8.** Potentiodynamic polarization of Fe-based amorphous coatings in HCl solutions with different content.

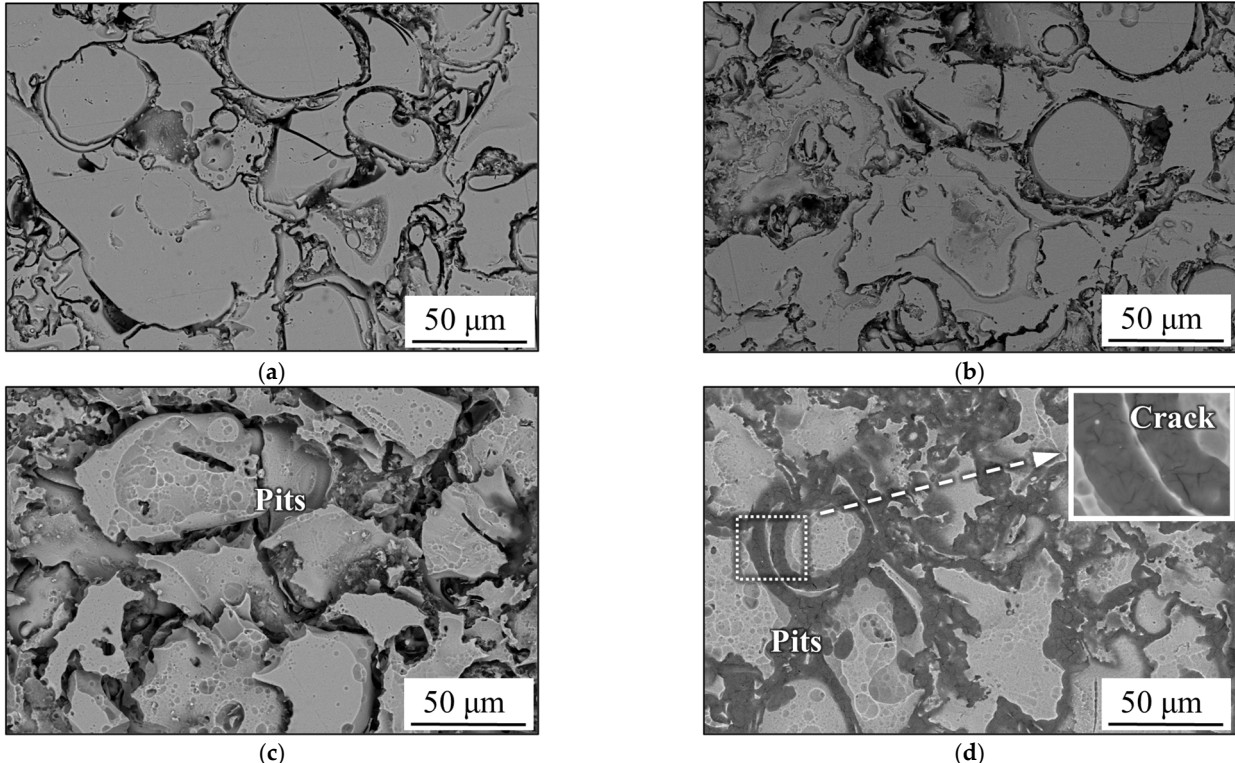

**Figure 9.** Corrosion morphologies of Fe-based amorphous coatings in HCl solutions, (**a**) 0.5 mol/L, (**b**) 1.0 mol/L, (**c**) 2.0 mol/L, (**d**) 4.0 mol/L.

### 3.3. Pitting Corrosion at the Interface

Oxygen could be detected by EDS with point scanning and the linear scanning model in the intersplat of the lamellar structure as shown in Figure 10, but not in the inner areas of the amorphous particles, such as point B in Figure 10a. This is because even if the ratio of kerosene to oxygen is theoretically capable of supporting complete combustion, the spray particles will inevitably contact with oxygen through the combustion chamber and the flame. The surface of the fused powder was oxidized to FeO at a high temperature, which matched the XRD results in Figure 4. When the oxygen was relatively sufficient at a low kerosene flow (23 L/h), or the heat accumulated per unit time and area at low travel speed of spray gun (4 m/min) was high, more FeO was precipitated due to the increased oxidation activity. With the increase in the kerosene flow, more oxygen was consumed, and the oxidation process was inhibited, but crystallization was initiated. Conversely, as

the gun moved faster (10 m/min), both the oxidation and crystallization were reduced because the accumulated heat could be released more quickly.

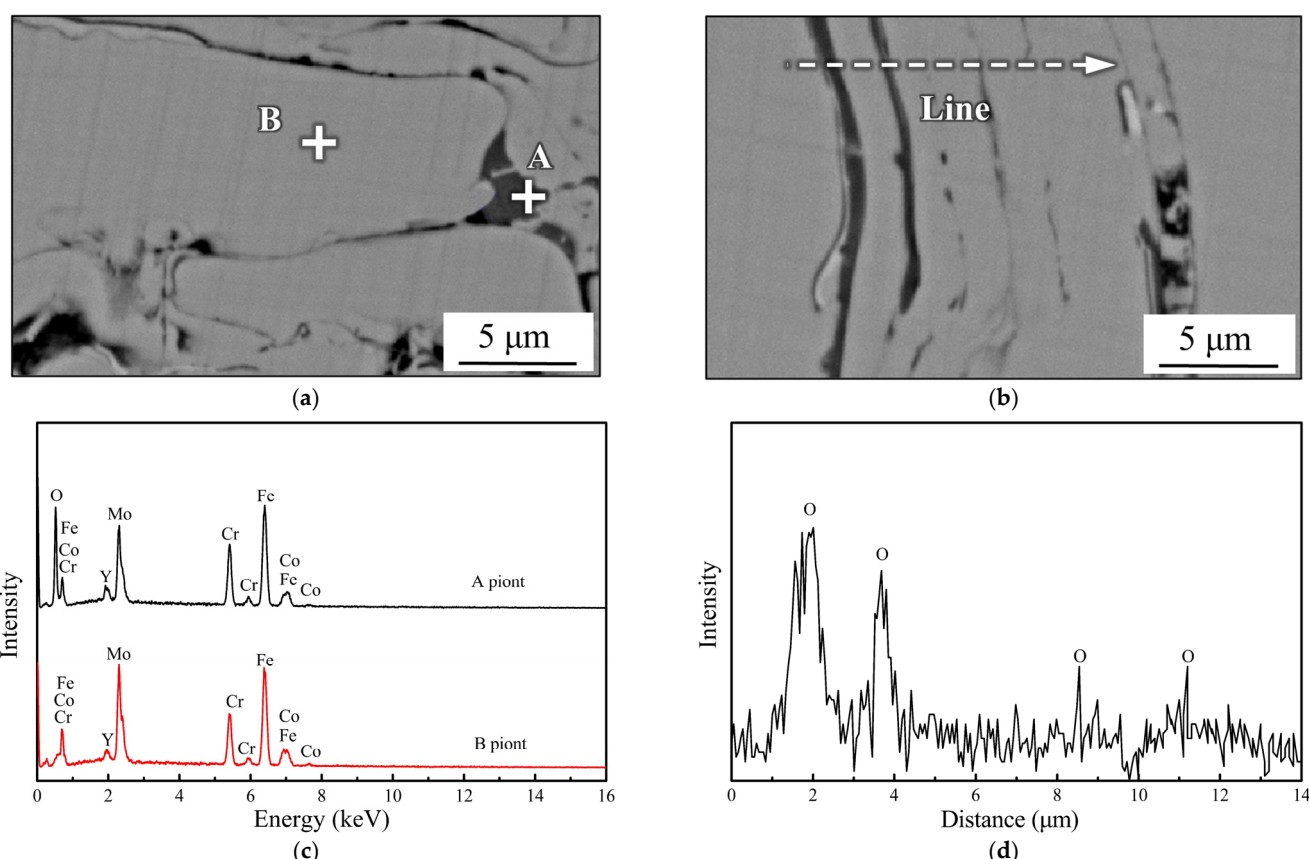

**Figure 10.** Interface characteristic of amorphous coating layer and elemental analysis with different scanning model, (**a**) and (**b**) lamellar structures, (**c**) point scanning point, (**d**) linear scanning.

Area A in Figure 11a was selected for the observation of the microstructure, and the bonding interface between the particles after thinning is shown in Figure 11b. The coating particles appeared to be in dark gray contrast while the bonding interface of the intersplat showed light gray contrast, suggesting that the form of the intersplat was metallurgical bonding rather than gaps. According to the SAED pattern of area B inside the amorphous particles, only a diffraction halo ring existed, indicating that the inner particles remained amorphous. At the same time, most of the metallurgical bonding area was also amorphous, but there were precipitated substances as shown in area D. It can be observed that nanocrystals were precipitated from amorphous components with the spots in the SAED pattern of area D representing crystalline $(Cr, Fe)_2O_3$, which is consistent with the PowderDiffraction File（PDF#02-1357）. Therefore, it could be inferred that the metallurgical bonding between the particles enhanced bonding as well as mechanical interlocking during spraying, and the metallurgical bonding interface of the intersplat was a mixed structure doped with nanocrystals.

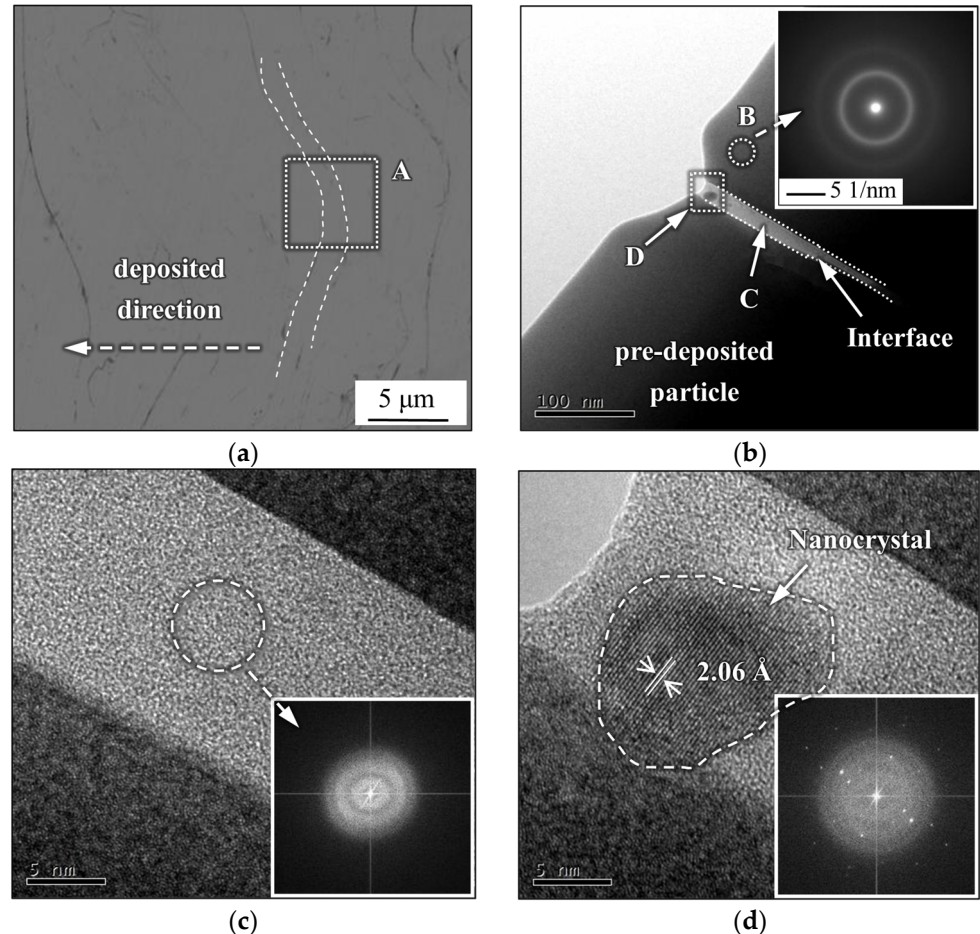

**Figure 11.** Interfacial microstructure of Fe-based amorphous coatings, (**a**) lamellar structure, (**b**) interfacial structure, (**c**) amorphous feature in C area, (**d**) nanocrystalline feature in D area.

To further study the initiation of pitting near the bonding interface, the sample was immersed in 6 mol/L NaCl solution for 2 h. In Figure 12, both the pre-deposited particles and the subsequent deposited particles still showed dark gray contrast, and the bright contrast associated with the bonding interface can be observed between them. It is worth noting that the area near the bonding interface on the side of the pre-deposited particles was slightly brighter than the interior of the particles, in which some pits were also distributed. Pitting was confirmed to occur near the particle bonding interface on the side of the pre-deposited particles. However, the same was not observed on the side of the subsequently deposited particles, suggesting a difference in composition near the surface of the particles during spraying, possibly due to active atomic diffusion.

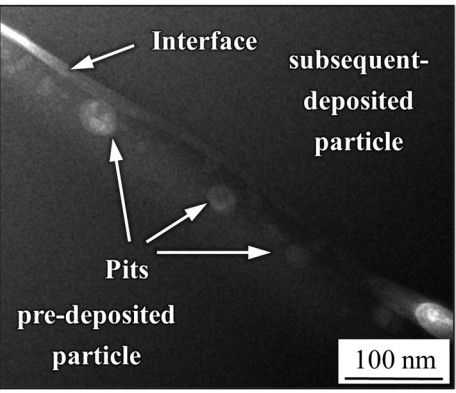

**Figure 12.** Pit corrosion near interface of Fe-based amorphous coatings in 6.0 mol/L NaCl solution.

## 4. Discussion

The combustion heat is primarily used to melt the powder and provide kinetic energy for high-velocity flight of spray particles. It is widely accepted that the accumulation of combustion heat increases the temperature of the combustion chamber and the pressure of the spraying flame as the kerosene flow increases. Therefore, under the condition of high heat input, the spray powder deposited on the substrate surface has a higher impact energy and degree of melting, which enables these particles to flow and spread more fully through plastic deformation before cooling and solidification [27]. At the same time, the full deformation of the particles fills up the pores between the powders, resulting in a higher density and more obvious lamellar structure that matches the characteristics of the coating cross-section morphology shown in Figure 2. The same conclusions were made in the study of thermally sprayed amorphous alloys [28,29] where the proportion of partially molten particles declines greatly, and the large-sized pores nearly disappear with the increase in heat input. However, the coatings prepared at different travel speed of the spray gun have similar lamellar structures, as shown in Figure 3. The degree of melting of the particles themselves does not change fundamentally because the travel speed only changes the quantity of powder deposited per unit time and area.

It is widely recognized that the mechanical interlocking is the main bonding mechanism of Fe-based amorphous HVOF coatings, which is caused by the wedging of the coating metal into the substrate metal [30]. If the sprayed powders gain more kinetic energy due to the increase in energy input in HVOF, the powders will have a higher degree of spreading and embedding when they impact the substrate or pre-deposited particles. Rough deposition interfaces, as shown in Figures 2 and 3, indirectly increase the area of mechanical interlocking and contact between particles [30]. At the same time, the oxide formed on the surface of spray particles during flight will also be crushed and follow the flow metal to disperse in the intersplats during the impact and the subsequent plastic deformation. Therefore, the high energy input facilitates mechanical interlocking and crushing of surface oxide, which increases the interlocking area and enables the inner amorphous components inside the particles to fully contact and form metallurgical bonding [31] (shown as Figure 11). However, too high thermal input will also lead to high surface temperature of the spray particles, which will stimulate oxidation and crystallization [32] as evidenced by the oxygen enrichment around the intersplats in Figure 10. To avoid excessive oxidation and crystallization, theoretically, the heat required for spraying is sufficient to melt the particles before cooling and forming. The spray energy input is expected to be converted to kinetic energy, which is also the application principle of the supersonic cold gas spray [33].

Based on earlier results [34,35], a schematic diagram of the amorphous coating formation for HVOF is shown in Figure 13. When the amorphous particle is heated in the combustion chamber and during the flying process, some oxygen elements will diffuse and remain along outer surface of amorphous particle in the high temperature, as shown in Figure 13a. When the heated and plasticized amorphous particles are deposited on the substrate or pre-deposited layer, the spherical particles will be deformed and flattened owing to the impact force, and a lamellar structure will be formed as shown in Figure 13b. At moment of impact, the outer oxidized film might be crushed and tend to flow with plasticized metals when extruded by the impact force as shown in Figure 13c. Therefore, the oxidized film will accumulate in the interfaces or cavities between the lamellar structures as observed in Figure 10. During the spraying process, the upper surface of lamellar structures is oxidized continuously until covered by next deposited layer. As a result, more heat accumulates on the surface of particles facing the flame, and atomic diffusion is more active. Since the oxides on the surface are in the form of Fe and Cr oxides [15], more Cr diffuses to the surface from the amorphous particles near the upper surface, resulting in Cr-deficient areas near the surface. Liu [16] also considered that a Cr-depleted zone about 100 nm wide existed in the area close to the intersplats, because Cr diffused to the particle surface due to the oxidation effect in the spraying process. Moreover, if the

spray particles with low energy are unable to impact the previous oxide layer and have sufficient plastic deformation, the cavities, and the thicker oxide layer around them will be preserved as shown in Figure 13e. Thus, the precipitation of nanocrystals and the distribution of Cr-deficient areas are consistent, that is, they appear near the upper surface oxidized layer of particles, as shown in Figure 13f.

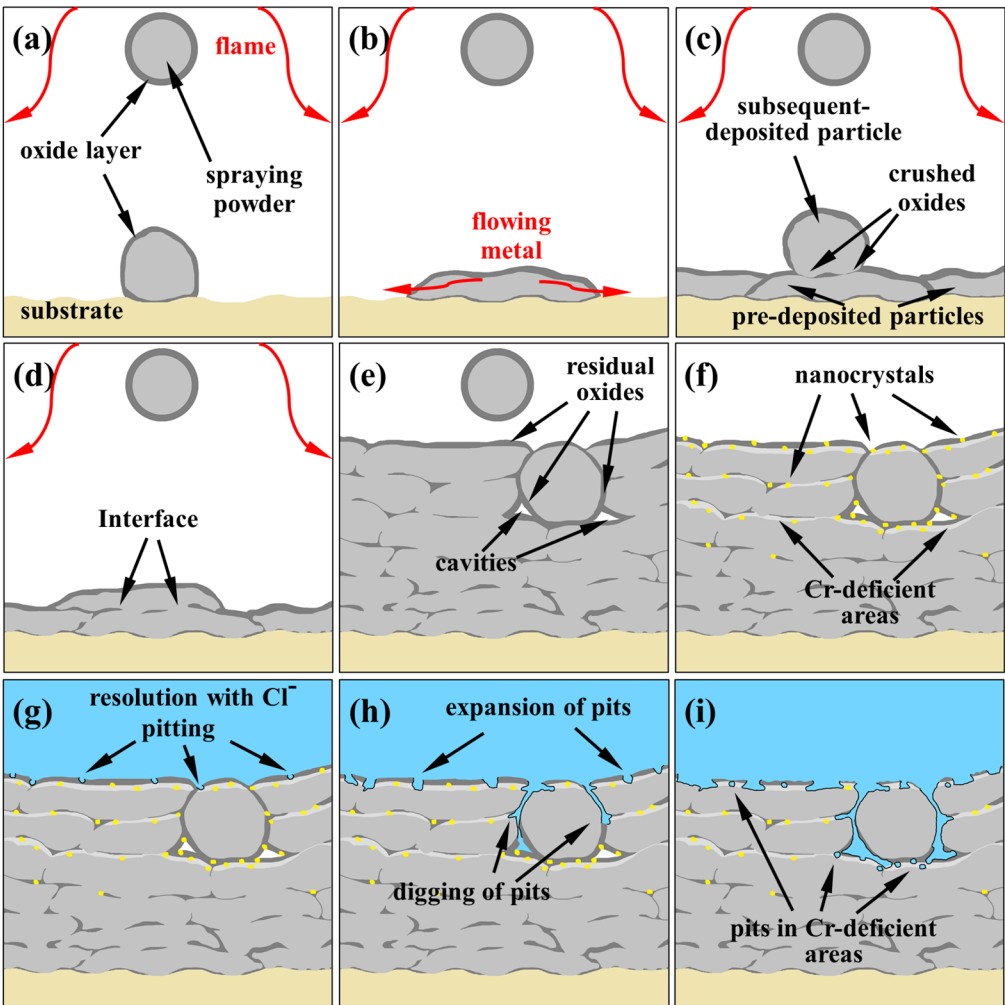

**Figure 13.** Schematic diagram of the deposition principle and the corrosion behavior of Fe-based amorphous coating: (**a**–**d**) spraying process, (**e**–**f**) the emergence of Cr-deficient areas, (**g**–**i**) corrosion behavior.

Fe-based amorphous and stainless-steel exhibit similar passivation behavior due to their inclusion of passivation elements. Passivation films with Fe and Cr oxides as the main components have higher corrosion potential in the corrosive environment. The Fe-based amorphous alloy has no grain boundaries and segregation of the typical defects in crystalline materials due to the uniformity of structure and composition. Therefore, compared with stainless steel, the Fe-based amorphous surface tends to form more uniform and dense passivation films with a high corrosion potential [36–38]. In HVOF spraying, insufficient heat will lead to poor coating formation, leaving more pores to destroy the structure uniformity of the coating (loose structure as shown in Figure 2a), while excessive heat will induce oxidation and crystallization of the coating and destroy the uniform composition of the coating [39–42]. It is generally believed that the nanocrystals on the coating surface form potential pitting sites after galvanic coupling with the passivation film [43]. Moreover, Cl⁻ tends to be enriched in the pores or pits, forming a concentration galvanic cell with the solution outside the pits. This is called the self-catalytic effect and accelerates

the formation and expansion of pits [44]. With the increase in the porosity of the amorphous coating layer, corrosion resistance in the chloride solution decreases notably [43,45].

Compared to the bulk specimen of the amorphous alloy, the corrosion of the HVOF coating is mainly dominated by pitting as pores and crystallization inevitably exist after spraying. A metastable pitting mechanism exists in the surface corrosion of Fe-based amorphous alloys [41], which means that the passivation film may repeatedly dissolve, break, and re-form at the corrosion site. It is generally believed that passivating elements play a crucial role in this process, among which Cr is the element that forms the passivating film directly and Mo can promote the re-formation of passivating film after dissolution. However, in the previous study [16], the Cr-deficient area was assumed to exist on both sides of intersplats, which cannot explain the fact that pitting occurs only on the side of the pre-deposited particles. Therefore, it is reasonable to speculate that the Cr-deficient areas are not symmetrical on both sides, because the heat distribution on both sides of the particles is not the same after the particles impact the substrate or the pre-deposited layers. On the side where the particle contacting with the pre-deposited layer, heat could be released quickly, while on the other side, heat continues to accumulate in the spraying flame. In other words, Cr diffusion to the surface and nanocrystal precipitation are more active on the side of the particles facing the flame, resulting in the formation of Cr-deficient areas and pitting initiation in Figures 11 and 12.

Therefore, the corrosion behaviors of Fe-based amorphous coating in the chloride solution can be summarized as in Figure 13g–i. Due to the heterogeneity of the microstructure and chemical composition in some areas, firstly, pit formation is initiated along the pre-deposited side of the interface, as shown in Figure 13g. As the pitting proceeds, the accumulation of chloride or hydrogen ion, which provokes the depassivation of the passive film and dissolution of the oxide layer [46], is promoted at the interface and in the cavities since the replenishment of the solution is not easy in the shallow areas. The concentration of chloride or hydrogen ion from the hydrolysis of metal ions increases gradually [47]. Moreover, the oxide contours formed between the lamellar particles can become the diffusion channels for the electrolyte, affecting the formation of the passive film and causing corrosion of the coatings [48]. Therefore, general corrosion will be accompanied near the interface, especially in the loose lamellar areas shown as Figure 13h. When a thick corrosion product is deposited on the surface, some 'dried riverbed' cracks occur on the surface with the internal stress release, as shown in Figure 9. As corrosion proceeds, when the outer thin layer of the lamellar splat or spherical particle is corroded, the remaining corroded particle will fall off from the amorphous coating, as shown in Figure 13i. Moreover, similar spalling was reported in the literature [49].

## 5. Conclusions

1.  Fe-based amorphous coating with high amorphous content and low porosity is obtained by high-velocity oxygen-fuel spraying (HVOF) using industrial raw materials; the amorphous volume fraction exceeds 99% and the porosity is lower than 1.0 %.
2.  The mechanical interlocking effect and metallurgical bonding act synergistically on the interface between the spray particles, and a model is proposed for the crystallization around the interface.
3.  Fe-based amorphous coatings exhibit superior corrosion resistance than SUS316 in the chloride solution.
4.  Pitting corrosion emerges along the pre-deposited side near the interface owing to the heterogeneity of the microstructure and chemical composition in certain areas, resulting from the nanocrystal precipitation and considerable Cr depletion.

**Author Contributions:** H.Z.: Investigation, Data curation, Formal analysis, writing—original draft. S.W. (Shanlin Wang): Conceptualization, Supervision, Writing—review & editing, Funding acquisition. H.L.: Methodology, Funding acquisition. S.W. (Shuaixing Wang): Resources. Y.C.: Supervision.

**Funding:** This work was supported by the National Natural Science Foundation of China (51965044) and (51971020).

**Institutional Review Board Statement:** Not applicable.

**Informed Consent Statement:** Not applicable.

**Data Availability Statement:** Not applicable.

**Conflicts of Interest:** The authors declare no conflict of interest.

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
