# Peer review of "Effect of Oxidation and Crystallization on Pitting Initiation Behavior of Fe-Based Amorphous Coatings"

_coatings, doi:10.3390/coatings12020176_

Round 1
Reviewer 1 Report
Remarks and questions:
p.1 praying - spraying?
Faermer – Farmer?
SAM2x5 SAM2X5 use the same marking
p.2 Density of fuel 0,825 - to add units
4 inches is not 101,6 cm but 10,16cm
- 4 the scale of picture 2 is missing
1) How can you explain the differences in the thickness of the coatings at different travel speeds (Fig. 3) when you write (p.13).
The degree of melting of the particles themselves does not change fundamentally because the travel speed only changes the quantity of powder deposited per unit time and area.
2) How can you explain the same values of the potential (Fig. 8) are in the solutions NaCl with different concentration when the potential is a function of material (it is the same) and environment (very different in concentration and aggressiveness)?
Reviewer 2 Report
Dear Authors,
Although your article is of high quality, there are some issues to be corrected.
- Please correct the scale superscriptions in Figure 1a. Make them all easily readable and in uniform font and size. Delete all wavy lines from the scale bars superscriptions in the article.
- Add scale bars to all microstructures in Figures 2 and 3.
- Correct the "A piont" and "B point" superscriptions in Figure 10.
- Correct the superscriptions in Figures 11 and 12 (remove unnecessary symbols)
- Please double-check the English in the article. For example, "When the amorphous particle is heated in the chamber and flying process, some oxygen will diffuse and remain along outer surface of
amorphous particle in the high temperature" - please correct the sentence. "as observe in Figure 10" - please correct to "as observed", etc. - Remove excess symbols from Figure 13
- I advise adding hardness measurements of the deposited coatings.
